

# The influence of spacetime curvature on quantum emission in optical analogues to gravity

**Maxime J. Jacquet[1,2]⋆ and Friedrich König[3]†**

**1** Faculty of Physics, University of Vienna, Boltzmanngasse 5, Vienna A-1090, Austria.
**2** Laboratoire Kastler Brossel, Sorbonne Université, CNRS,
ENS-Université PSL, Collège de France, Paris 75005, France
**3** School of Physics and Astronomy, SUPA, University of St. Andrews,
North Haugh, St. Andrews, KY16 9SS, United Kingdom

⋆ maxime.jacquet@lkb.upmc.fr, † fewk@st-andrews.ac.uk

## Abstract

Quantum fluctuations on curved spacetimes cause the emission of pairs of particles from the quantum vacuum, as in the Hawking effect from black holes. We use an optical analogue to gravity to investigate the influence of the curvature on quantum emission. Due to dispersion, the spacetime curvature varies with frequency here. We analytically calculate for all frequencies the particle flux, correlations and entanglement. We find that horizons increase the flux with a characteristic spectral shape. The photon number correlations transition from multi- to two-mode, with close to maximal entanglement. The quantum state is a diagnostic for the mode conversion in laboratory tests of quantum field theory on curved spacetimes.

**Numerical data repository** F. Koenig and M. J. Jacquet, 2020, *The influence of spacetime curvature on quantum emission in optical analogues to gravity*, Dataset, University of St Andrews Research Portal (2020), doi:10.17630/cbf5b4f6-2c82-4eb5-9aaf-b596bf8090d8.

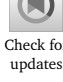

# 1 Introduction

In a curved spacetime, quantum fluctuations lead to the spontaneous emission of particles. Famously, if the curved spacetime contains an event horizon, it is predicted to emit pairs of particles via the Hawking effect [1, 2]. However, the (static) black hole event horizon is not the only 'regime of spacetime curvature' that leads to particle emission. Analogue spacetimes are effective wave media that allow for table-top experiments on configurable curved spacetimes [3]. In addition to static black holes [4–10], it is also possible to create *e.g.* (static) white hole event horizons [4,6,8,11–15], rotating geometries analogous to Kerr black holes [16,17], expanding universes [18–20] or even (static) two-horizon interactions [21,22]. For those systems featuring a static horizon, the classical frequency shifting of waves at the horizon has been the traditional benchmark to demonstrate analogue gravity physics, although scattering of waves that could not be associated with a horizon has also been observed [6,11,13,23,24]. Correlated pairs of particles from horizons are considered an unmistakable signature of the quantum Hawking effect [26,27], and have therefore been extensively studied for fluid systems, in which their entanglement in various dispersive regimes has been investigated [28–37]. However, these studies have not contrasted horizon and horizonless spontaneous emission, and neither has this been done in other analogue systems and with many modes. Ergo, the question of the influence of the spacetime curvature on quantum emission in analogues to gravity arises: *e.g.* what differentiates emission at the horizon (the Hawking effect) from horizonless emission?

In this letter, we demonstrate the transition between different 'regimes of spacetime curvature' using a dispersive analogue optical system [4, 6, 8, 12, 38–40]. Due to dispersion, each frequency mode experiences different kinematics as of a spacetime with or without horizons. To pinpoint the matter further, we use a system in which particles are emitted from a single point: an approximately step-shaped optical pulse moves through a dispersive medium, which we consider in 1D. The pulse intensity increases the refractive index $n$ of the medium by the optical Kerr effect, creating a moving refractive index front (RIF). Light under the step is slowed by the increased index, *i.e.*, light of some frequencies will be slowed below the pulse speed and captured into the RIF. This is in analogy with the kinematics of waves around a black-hole event horizon [3, 41, 42]. At other frequencies, however, light follows different kinematic scenarios (*i.e.*, trajectories of waves). Thus this simple optical system allows us to contrast the quantum emission in these different scenarios. Moreover, analytical solutions to the scattering exist.

We present all the possible kinematic scenarios for modes at the RIF and thus explain how the interplay between the step height (the magnitude in the index change) and the dispersion of the system gives rise to distinct regimes of spacetime curvature. We then use an analytical method developed in [43,44] to describe the scattering of modes at the RIF and calculate the spontaneous emission in each regime of spacetime curvature. Furthermore, we quantify the bipartite entanglement of modes using the logarithmic negativity, which is an entanglement monotone. Spectra of entanglement of key modes indicate multimode entanglement which is highly dependent on the kinematic scenario. Thus we complete the entanglement measure with the degree of entanglement calculated between all mode pairs in all regimes of spacetime curvature.

# 2 Regimes of spacetime curvature

The metric of spacetime in the vicinity of a Schwarzschild black hole can be expressed in the Painlevé-Gullstrand coordinates [45, 46]: in 1+1D its line element is

$ds^2 = -(c^2 - \beta^2)dt^2 + d\zeta^2 + 2\beta \, dt \, d\zeta$, with the Newtonian escape velocity $\beta = (2Gm/\zeta)^{1/2}$, $G$ the gravitational constant, $m$ the mass of the black hole, $\zeta$ the spatial coordinate, $t$ the proper time, and $c$ the speed of light in vacuum [47–49]. In this so-called "River Model" of the black hole, the metric describes ordinary flat space (and curved time), with space itself flowing towards $\zeta = 0$ (the spacetime singularity) at increasing velocity $\beta$ [50, 51]. This acceleration of the flow velocity of space towards $\zeta = 0$ is the manifestation of the curvature of spacetime around the black hole: before the horizon, the flow velocity is subluminal, $\beta = c$ at the horizon (when $\zeta = \zeta_{Schw}$, the Schwarzschild radius) and the flow velocity of space is superluminal inside the horizon. The kinematics of waves propagating on this spacetime is determined by the curvature: for $\zeta > \zeta_{Schw}$, motion is possible towards and away from the horizon, whereas for $\zeta < \zeta_{Schw}$, motion is restricted from the horizon towards the singularity. Note that, upon a mere time-reversal of the metric, we obtain another regime of spacetime curvature — the white hole. Here, the kinematics of waves are such that motion in the interior region may only be directed from the singularity towards the horizon, whereas two-way motion is again possible in the outside region. That is, the white hole horizon prevents waves from entering the inside region.

Via the River Model, we understand the event horizon of the black (or white) hole as the interface between regions of sub- and superluminal space flow. In our optical analogue, such an interface is created by a step change in the refractive index of the medium in which light propagates. This RIF is illustrated in Figure 1 **a** in co-moving frame coordinates $x$ and $t$.

The RIF separates two regions of homogeneous refractive index whose dispersion is modelled by the generic Sellmeier dispersion relation [52]

$$c^2 k^2 = \omega^2 + \sum_{i=1}^{3} \frac{4\pi \kappa_i \gamma^2 (\omega + uk)^2}{1 - \frac{\gamma^2 (\omega + uk)^2}{\Omega_i^2}}, \tag{1}$$

with $\omega$ and $k$ the frequency and wavenumber in the frame co-moving with the RIF at speed $u$ ($\gamma = [1 - u^2/c^2]^{-1/2}$). $\Omega_i$ and $\kappa_i$ are the medium resonant frequencies and elastic constants. The refractive index of most dielectrics is sufficiently well described by 3 resonances. The change in index, $\delta n$, at $x = 0$ between the two regions is modelled by a change in $\kappa_i$ and $\Omega_i$ in (1). As illustrated in Fig.1, $\delta n$ manifests itself by a change in the dispersion relation between the low ($x > 0$, black curve) and high ($x < 0$, orange curve) refractive index regions. For a single frequency $\omega$, the eight solutions of the polynomial (1) define eight discrete wavenumbers $k$: the eight 'modes' of the field. The Klein-Gordon norm of a mode is a constant of the motion, and is positive (negative) if the laboratory frame frequency $\Omega = \gamma(\omega + uk)$ is positive (negative) [44, 53]. In figures 1 and 2, we focus on the optical frequency modes only. The detailed treatment [44] accounts for modes of all laboratory frame frequencies.

The dispersion curves in Fig.1 have the negative- (positive-) norm modes on the left (right). For all $\omega$, there is one negative-norm mode and either three positive norm modes, or one positive norm mode.[1] One of the three positive-norm modes is the only mode solution of (1) with positive group velocity $\frac{\partial \omega}{\partial k}$. We call it 'mid-optical' — 'moL' on the left, and 'moR' on the right. There are also the 'low optical' mode $lo$, the 'upper optical' mode $uo$, and the 'negative optical' mode $no$. $lo$, $uo$ and $no$ all have negative group velocity. Thus at a frequency $\omega$ with three positive norm modes ($k$'s), light may propagate in two directions: towards and away from the interface at $x = 0$. These kinematics are characteristic of a spacetime curvature with a subluminal space flow. We refer to these intervals as *subluminal intervals*: $[\omega_{minL}, \omega_{maxL}]$ and $[\omega_{minR}, \omega_{maxR}]$. Motion is restricted to the negative $x$ direction in all other frequency intervals, which we call *superluminal intervals*. Only in scenarios **b** and **d** is a subluminal

---

[1]At frequencies with only one positive norm mode, the two other mode solutions of Eq.(1) have become complex.

Figure 1: Possible kinematic scenarios at the RIF. Time (left) and frequency (right) illustrations of propagating modes are shown for different comoving frequencies $\omega$ (**a-d**). The black (orange) dispersion curve corresponds to the low (high) index region. Single-frequency modes of $\omega$ are identified by intersections with the blue dashed line. Possible kinematic scenarios, analogous to changes of space-time curvature: **a**, horizonless scenario ($\omega < \omega_{minL}$); **b**, white hole scenario ($\omega_{minL} < \omega < \omega_{minR}$); **c**, horizonless scenario ($\omega_{minR} < \omega < \omega_{maxL}$); **d**, black hole scenario ($\omega_{maxL} < \omega < \omega_{maxR}$).

region paired with a superluminal region, creating a horizon: in **b** (**d**) the superluminal flow is towards (away from) the horizon as in a white (black) hole. On the other side of the horizon the flow is subluminal. *E.g.* in Figure 1 **d** a positive norm mode (*moR*) — Hawking radiation — allows for energy to propagate away from the hole in the subluminal region, while its negative norm partner (*nol*) falls inside the horizon.

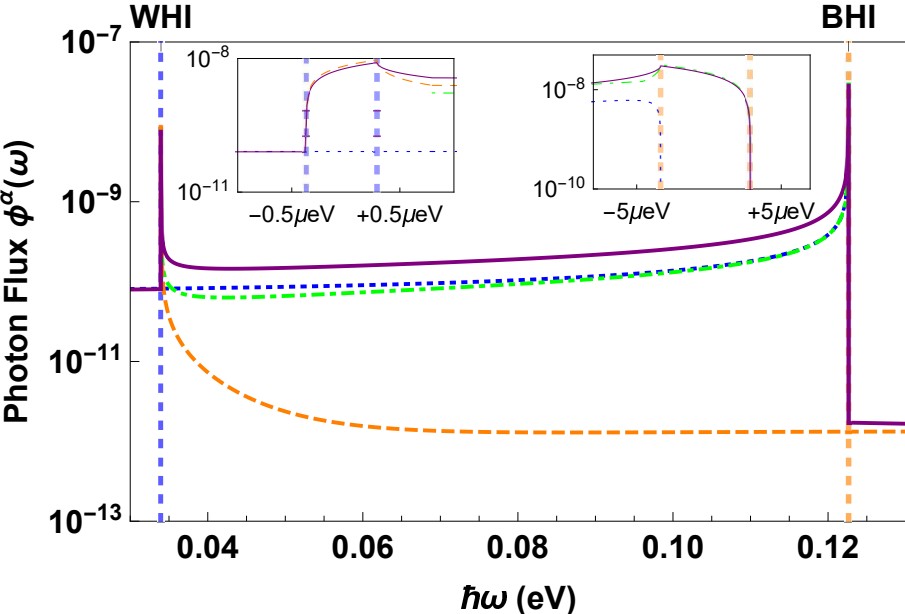

Figure 2: Spectra of photon flux (2) of the optical modes in the moving frame: *noL*, purple solid line; *uoL*, blue dotted line; *moR*, green dot-dashed line; *loL*, orange dashed line. Insets zoomed-in around the centre frequency of the WHI and BHI.

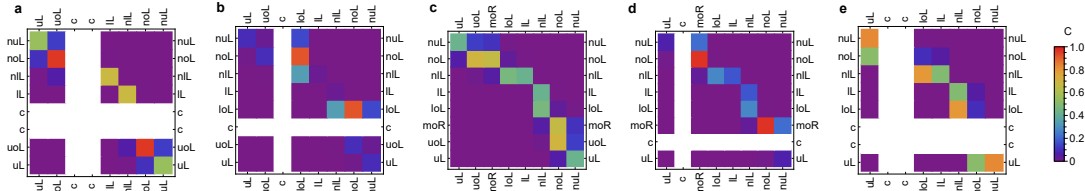

Figure 3: Photon number correlations between the 8 outgoing modes for five typifying frequencies: **a-d** as in Fig.1 and **e** for high $\omega$; $c$ — a complex mode; *ul, nl, ll, nul* — non-optical modes.

This analysis shows that optical analogues, unlike other analogue systems (see Appendix A), give access to regimes of spacetime curvature with both sub- and superluminal flows — that is, to event horizons — as well as only subluminal or only superluminal flows. For example, the kinematics over the horizonless frequency interval (Fig.1 **c**) resemble a change of curvature of spacetime induced by a gravitational wave, a wiggle in spacetime. As a result, considering all the four kinematic scenarios (Fig.1 **a-d**) in a single system enables the comparison of the emission with and without horizons.

We describe the transformation of ingoing modes into outgoing modes by the scattering matrix formalism and calculate the paired emission in all regimes of curvature. Our method to calculate the scattering matrix $S(\omega)$ [44] allows us to analytically describe the mode coupling and takes all modes into account in all possible kinematic scenarios for any RIF height, *i.e*, refractive index change $\delta n$.

## 3 Quantum emission in all regimes of spacetime curvature

We calculate spectra and mode-correlation maps to study the influence of the change in spacetime curvature on quantum emission. We present the properties of quantum emission as meas-

ured in the frame co-moving with the RIF, where the frequency $\omega$ is conserved. We emphasise that the spectra herein are directly comparable with those obtained for laboratory-frame observations in 'fluid' systems.[2]

We compute the spontaneous photon flux in the moving frame [44]:

$$\phi^\alpha(\omega) = \frac{1}{2\pi} \sum_{\beta \notin \{\alpha\}} \left| S_{\alpha\beta}(\omega) \right|^2. \tag{2}$$

Here, $\{\alpha\}$ is the set of modes that have a positive (negative) norm if $\alpha$ is of positive (negative) norm. The photon flux results from the scattering of *in* modes into *out* modes of opposite norm. All *in* modes are in the vacuum state. In this paper, we consider the example of a RIF of height $\delta n = 2 \times 10^{-6}$, moving at $u = {}^2\!/\!{}_3 c$ in bulk fused silica [44].

As can be seen in Fig.2, the spontaneous emission flux (2) peaks in two narrow frequency intervals. The low- and high-frequency intervals, white hole interval (WHI) and black hole interval (BHI), correspond to white and black hole emission, respectively. Over the horizon intervals (insets), the spectrum has a characteristic 'shark fin' shape: on one side the overall emission cuts off by many orders of magnitude as the emitting Hawking mode (*loL* or *moR*) ceases to exist. On the other side we enter kinematic scenario **c** (in Fig.1), leading to an abrupt decrease in emission. Outside these intervals, the emission decreases to a near constant level. All photons produced have partners of opposite norm and the emission follows the kinematics explored in Fig.1: for example, over the WHI, emission is mainly into modes *noL* (purple line) and *loL* (orange-dashed line). Over the BHI, emission is strongest into modes *noL* and *moR* (green-dot-dashed line). Because *noL* is the only negative norm mode of optical frequency, the flux in this mode is high.

The partnered emission is a key signature in a laboratory experiment. In order to separate the kinematic scenarios and in view of the large bandwidth of emission, photon number correlations in the spectral density have to be measured with frequency resolving detectors. In [44], we calculate the photon-flux Pearson correlation coefficient between detectors 1 and 2 of bandwidth $\Delta_1$ and $\Delta_2$ in the moving frame, corresponding to modes $\alpha$ and $\alpha'$ (including non-optical modes), as

$$C(\hat{N}_1^\alpha, \hat{N}_2^{\alpha'}) = \frac{\Delta^2}{\Delta_1 \Delta_2} \frac{\left| \sum_{\beta \notin \{\alpha\}} S_{\alpha\beta}^* S_{\alpha'\beta} \right|^2}{(\nu^\alpha \nu^{\alpha'})^{1/2}}. \tag{3}$$

Here, $\hat{N}_1^\alpha$, $\hat{N}_2^{\alpha'}$, respectively, is the photon number operator at detector 1 and 2, $\Delta$ is the frequency interval detected by both detectors and $\nu^\alpha = 2\pi \phi^\alpha (2\pi \phi^\alpha + 1)$. In (3) the notation is omitting dependencies on $\omega$.

Fig.3 shows correlations between all outgoing modes for typifying frequencies $\omega$ and $\Delta_1 \Delta_2 = \Delta^2$.[3] Strongest correlations (red) can be found between optical modes, which are also the strongest emitters. Because *noL* is the unique negative-norm optical mode, we find strong correlations between this mode and positive-norm optical modes. The correlation coefficients are different if horizons exist (Fig.3 **b**, **d**). Over the WHI and BHI, there is only a single large correlation between modes *noL* and *loL* ($C(N^{noL}, N^{loL}) = 0.97$), and *noL* and *moR* ($C(N^{noL}, N^{moR}) = 0.92$), respectively. With horizons, the most strongly correlated pairs of modes correspond to the Hawking emission mode and the partner mode.

The correlation structure is very different when there are no horizons. In the low frequency interval (Fig.3 **a**), significant correlations exist between three mode pairs simultaneously:

---

[2]Note that the reference frames of optical and 'fluid' analogues [9, 11, 13–15] are exchanged: the rest frame of the moving fluid corresponds to the moving frame of the optical experiment, and vice versa.

[3]Whilst, seen from the moving frame, the detectors detect at the same frequency, in the laboratory frame they detect at different frequencies, see [44].

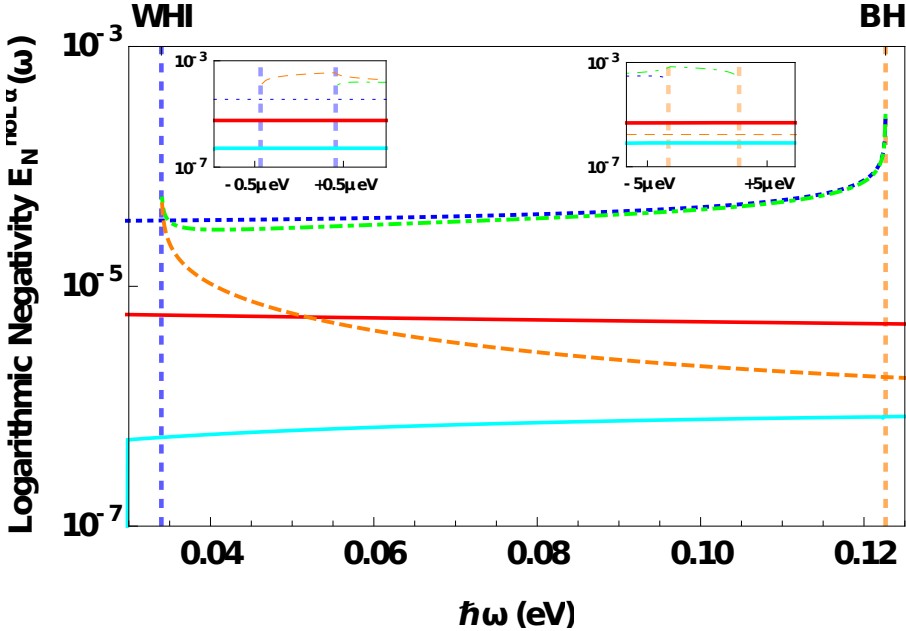

Figure 4: Entanglement of *noL* with the positive norm modes. *lL*, turquoise solid line; *loL*, orange dashed line; *moR*, green dot-dashed line; *uoL*, blue dotted line; *uL*, red solide line. Insets zoomed-in around the center frequency of the WHI and BHI. $\alpha$ a positive norm mode.

$C(N^{nuL}, N^{uL}) = 0.56$, $C(N^{noL}, N^{uoL}) = 0.97$ and $C(N^{nlL}, N^{lL}) = 0.68$. In the middle frequency interval (Fig.3 **c**), significant correlations exist between five modes pairs: $C(N^{nuL}, N^{uL}) = 0.43$, $C(N^{noL}, N^{uoL}) = 0.71$, $C(N^{noL}, N^{moR}) = 0.67$, $C(N^{nlL}, N^{loL}) = 0.47$ and $C(N^{nlL}, N^{lL}) = 0.43$. In the high frequency interval (Fig.3 **e)**), significant correlations exist between four mode pairs: $C(N^{nuL}, N^{uL}) = 0.83$, $C(N^{noL}, N^{uL}) = 0.52$, $C(N^{nlL}, N^{loL}) = 0.80$, and $C(N^{nlL}, N^{lL}) = 0.50$. Without horizons, strong correlations exist albeit with more than two modes.

In addition to the observations above, we note that strong correlations are possible without horizons (*e.g.* in Fig.3. **a**). Therefore, horizons lead to strong correlations but the converse is not true. Clearly, if we considered a two-mode system without horizons, any emission would be in correlated pairs with unit correlation coefficient.

To summarise, the flux of spontaneous emission is dominated by white- or black hole-horizon physics and drops significantly beyond that. Hence, the spectral characteristic is a 'shark fin' shape. Over the analogue white- and black hole intervals, paired two-mode emission at optical frequencies dominates.

## 4 Horizon condition and entanglement

Given the ample photon number correlations of Fig.3, we now proceed to assess the entanglement in the output state. The two-mode correlation coefficient does not reach unity at any frequency because more than two modes are involved in the correlation. As a result, although the entire output state is pure, the two-mode output will always be partially mixed, and coupling to modes outside the optical branch is responsible for this.

Our measure of entanglement is the logarithmic negativity (LN) $E_{\mathcal{N}}$ [54].[4] Contrarily to the Peres-Horodecki criterion used in fluid systems [29–32, 34, 35, 37] to determine whether

---

[4]Jacquet M, König F. Further details on the calculation to be published elsewhere

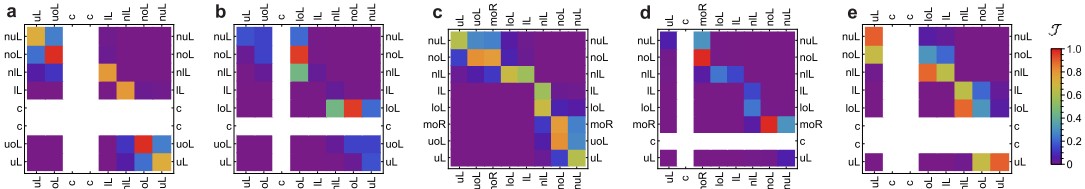

Figure 5: Degree of entanglement (4) between the 8 outgoing modes for the five typifying frequencies of Fig.3. Here $c$ indicates a complex mode; $ul, nl, ll, nul$ are the non-optical modes.

the state at the output is nonseparable, the LN is a measure of entanglement [55] for arbitrary bi-partite states under LOCC [56]. In addition, we normalise the LN to the 'degree of entanglement', which enables us for the first time to contrast the degree of entanglement of the output two-mode state of horizon and horizonless emission.

We calculate the LN between two output modes, tracing over the remaining modes. In particular, we are interested in the only negative norm optical mode, $noL$, because all other optical modes are coupling mainly to this mode. In Fig. 4 the spectrum of the LN is shown for $noL$ and each other positive norm mode. Similar to the emission spectra, the entanglement is peaked at both horizon intervals (WHI, BHI). Thus horizons efficiently entangle the light. However, $E_\mathcal{N}$ is an extensive quantity that increases with emission strength. In order to evaluate how strongly the photons of the two-mode state at the output are entangled, we compute the degree of entanglement $\mathcal{J}$. We define $\mathcal{J}$ as the LN ($E_\mathcal{N}^{\alpha_1\alpha_2}$) relative to the LN of a maximally entangled two-mode state of the same energy [57].[4]

$$\mathcal{J} = \frac{E_\mathcal{N}^{\alpha_1\alpha_2}}{4\operatorname{arsinh}\left(\sqrt{\frac{\phi^{\alpha_1}+\phi^{\alpha_2}}{2}}\right)}, \tag{4}$$

In Fig.5, we plot the degree of entanglement for the same typifying frequencies as in Fig.3. The degree of entanglement is generally strongest between optical modes. In particular, the positive-norm optical modes $moR$ and $loL$ are close to being maximally entangled with mode $noL$ over the black- and white-hole intervals, respectively (**b**, **d**). In other words, with horizons the Hawking mode and partner is in a close to maximally entangled, pure state. Without horizons (Fig.5 **a**, **c**, **e**), the degree of entanglement decreases for all mode pairs (except between modes $uoL$ and $noL$ in Fig.5 **a**) and the two modes are in a mixed state.

The similarity of Figs.3 and 5 may suggest that the entanglement can be inferred from the photon-number correlations. In Fig.6, we plot the degree of two-mode entanglement (4) against the corresponding correlation coefficient (3) for each mode pair at each of the five typifying frequencies of Fig.3. We observe that $\mathcal{J}$ is not a function of the correlation coefficient $C$. In other words, there is no unique, one-to-one relation between the correlation coefficient and the degree of two-mode entanglement as measured via the logarithmic negativity. States of different degree of mixedness may exhibit the same correlation coefficient. Furthermore, we remark firstly that for a finite correlation coefficient the degree of entanglement is limited below unity. Secondly, correlations close to unity indicate close to maximal entanglement. The degree of entanglement obtained for $C < 0.5$ varies largely, whereas for $C > 0.5$ the degree of entanglement can be reliably inferred from the correlations. Lastly, even small correlation coefficients indicate some entanglement. Further investigations of the relation between $\mathcal{J}$ and $C$ beyond this particular analogue are needed. It would be interesting to verify whether the non-uniqueness of the correspondence between the photon-number correlations and the degree of entanglement is specific to the measure of entanglement chosen — we leave these considerations to future investigations.

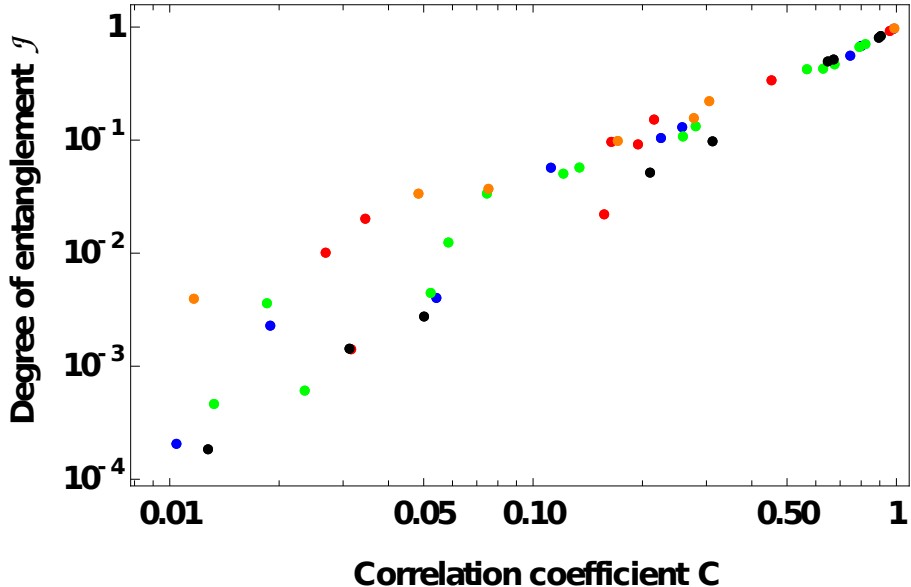

Figure 6: Relation between the degree of entanglement (4) and the photon-number correlations (3) for the 5 typifying frequencies of Fig.3 (**a**-**e**). Blue — low $\omega$ (cf. **a**); red — WHI (cf. **b**); green — medium $\omega$ (cf. **c**); orange — BHI (cf. **d**); black — high $\omega$ (cf. **e**). $\alpha$ a positive norm mode.

## 5 Conclusion

We showed how strong dispersion can lead to novel multimode quantum dynamics stemming from the relation between horizon and horizonless emission in gravity analogues. Our work sheds light on the interplay between kinematics and parametric amplification in optics. The kinematic aspects of a physical system can thus be used to define quantum information processing tasks in a gravitational context.

We observed a striking transition in the quantum statistics when going from one kinematic scenario to the other, which we expect to be ubiquitous to gravity analogues, from optics through fluids to solid state. The influence of the kinematic scenario or regime of spacetime curvature is directly imprinted onto the quantum state. Hence we observed the entanglement and mixedness of any two-mode state to directly depend on it. These effects are largely robust against changes in dispersion and in the magnitude of the refractive index change [44].[4] For our analysis we calculated the two-mode spectra as well as the logarithmic negativity of an optical analogue for the first time. This way we have put limits on the entanglement of an analogue system due to coupling to modes other than the Hawking pair. Particle number correlations and the degree of entanglement are key in characterising a variety of effects in all analogue gravity experiments such as cosmological pair creation by passing gravitational waves or expanding/contracting universes [19, 20, 28, 58], the so-called black hole laser [59], analogue wormholes, and the quasi-bound states of black holes [17, 60]. Importantly, the multimode analysis allows us to put limits on the thermality of the output state of spontaneous, quantum emission for the first time: since the degree of entanglement stays below unity, the state is not completely thermal.[5] In this way, our work demonstrates that the quantum state is a diagnostic for mode conversion on curved spacetimes.

---

[5]Note that this is different from the considerations drawn in, *e.g.* [34, 35, 61] on the influence of the grey-body factor on the thermal shape of the spectrum.

## Acknowledgements

The authors are thankful to Bill Unruh, Matthew Thornton, Viktor Nordgren, Mathieu Isoard and Valeria Saggio for insightful conversations.

**Author contributions**  Both authors contributed equally to the work and the writing of the manuscript.

**Funding information**  This work was funded by the EPSRC via Grant No. EP/L505079/1 and the IMPP.

## A   Dispersion and spacetime flow in a BEC

Here we briefly comment on the kinematic scenarios realised in typical experimental conditions in a BEC [9]. Similar to the RIF studied in the present paper, an analogue gravity experiment in a BEC is realised by two regions in the so-called 'waterfall' configuration: a high potential region and a low potential region with a step-like transition (see *e.g.* [62]), defining a high and low atomic density region, respectively.

In the moving frame — in which the fluid is at rest — the dispersion relation is of the generic form [9,37,63]

$$\Omega^2(K) = K^2\left(1 + \frac{K^2}{4}\right),\tag{5}$$

with $\Omega$ the moving frame frequency and and $K$ the wavenumber. The frequency in a frame moving with velocity $u$ is related to the laboratory frame frequency $\omega$ (in which the step is at rest and in which the measurements are performed) by $\Omega = \omega + uK$. The dispersion in the laboratory frame is shown in Fig.7, with the high potential region on the left, and the low potential region on the right.[6] The negative norm branch is shown in red, and the positive norm branch in black.

The speed of the BEC flow is larger in the low potential region than it is in the high potential region. Thus, the shape of the dispersion relation changes (by the Doppler effect) between the two regions. In particular, part of the negative norm branch (at $K > 0$) in the low-potential (high flow velocity) region is "pulled up" to positive frequencies.

Exactly as in our analysis [44] (but with the frames exchanged), the modes are found at the intersection points between a contour of positive $\omega$ and the various branches of the dispersion relation. In the high-potential (low flow velocity) region, there are only ever two modes: they have a positive norm and, from low to large $K$, one has negative group velocity (it moves away from the interface) while the other has positive group velocity. After *e.g.* [33,37,64], we call these modes 'u|out' and 'u|in', respectively. In the low-potential (high flow velocity) region, the number of modes varies as a function of frequency. From $\omega = 0$ to $\omega = \omega_{threshold}$, there are two positive-norm modes and two negative norm modes. From low to large $K$, we find first the two positive-norm modes — 'd1|in' (negative group velocity, moves towards the interface) and 'd1|out' (positive group velocity, moves away from the interface) — and then the two negative-norm modes — 'd2|out' (positive group velocity, moves away from the interface) and 'd2|in' (negative group velocity, moves towards the interface). For larger frequencies, only the two positive-norm modes remain.

If one focuses the analysis on the kinematics of modes 'u|out' and 'd2|out' at low $k$, the dispersion is approximately linear and the flow velocity is subsonic in the left region and su-

---

[6]This is a typical dispersion relation from the BEC literature, see *e.g.* [33].



personic in the right region. The mode analysis above shows, when input and output modes are taken into account, that two-way motion is possible in both regions of the BEC.

Our analysis of the optical analogue demonstrates that all modes (optical as well as non-optical) significantly modify the quantum state away from the close-to two-mode squeezed vacuum in particular when two-way motion is possible in both regions. Future work should apply this analysis to the BEC analogue also.

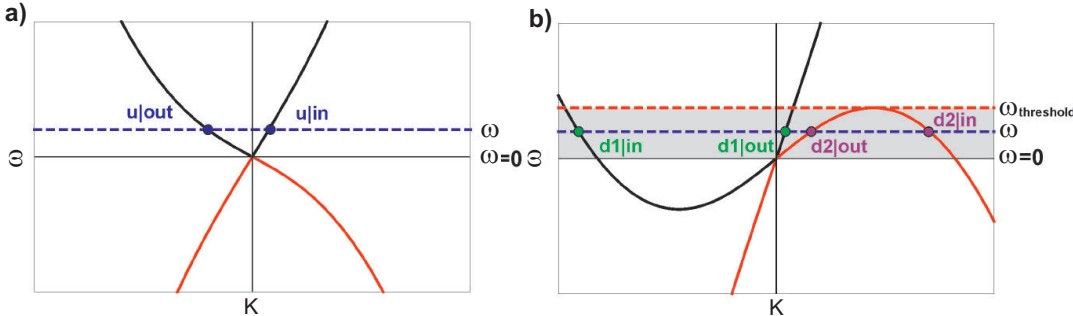

Figure 7: Dispersion relation in the laboratory frame of a BEC analogue: **a)** high-potential (low flow velocity) region; **b** low-potential (high flow velocity) region. Black — positive-norm branch; Red — negative-norm branch.

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
