# Peer review of "The influence of spacetime curvature on quantum emission in optical analogues to gravity"

_SciPost Physics, doi:SciPost Phys. Core 3, 005 (2020)_

## Round 1 · Referee Report · Anonymous (Referee 1) · 2020-2-25

Strengths

1- new results on entanglement in analogue systems
2- timely

Weaknesses

1- the paper is too concise, it would benefit from a more pedagogical presentation

Report

This paper is a theoretical study of an optical analogue of a curved space-time. This analogy is implemented by a moving step-shaped region of increased refractive index in a one-dimensional optical medium. Depending of the frequency window considered the system may display a black hole or a white hole horizon, or also no horizon at all.

This type of configuration has already been studied experimentally and theoretically in Refs. [4,11,42,45] of the paper, and by the authors themselves (Ref. [46]). Due to the complexity of the Sellmeir dispersion relation, the phenomenology of the system is complex, and the authors wisely spend sometime to present the different possible regimes (section 2).

For later use, the authors also need to explain how they compute the quantum emission: this is briefly done in Section 3 where, if I am not mistaken, the S matrix is used [Eqs. (2) and (3)] without having been preliminary nor subsequently introduced. Actually the S matrix appears already in section 2, but with a different notation, making the connection with Eqs. (2) and (3) difficult for the non-expert reader (note also that 3 different notations are used in the 3 different places where the S matrix appears). This section would be more easily understood if extended.

Section 4 is the most important one of the paper. It presents a qualitative study of the bipartite entanglement using the logarithmic negativity. Whereas previous studies in the domain used Peres-Horodecki criterion or the violation of Cauchy-Schwarz inequalities as signatures of non-separability, to my knowledge this is the first time that an entanglement monotone is used in the field of analogue physics. I was also quite interested by the remark of appendix B underlying the connection between correlation and entanglement. It is a pity that the most important results of the paper are presented in less that a page, with details postponed to a latter publication (footnote 4).

I believe that this work contains important new results on a subject presently of interest, both at the experimental and the theoretical point of view. However, this paper seems to have been drafted as a letter, probably initially designed for another journal than SciPost. I am not sure that its present format meets the editorial criteria of SciPost Physics. I also think that it would benefit of a more detailed presentation (secs. 3 and mostly section 4). Once the pedagogical level of this work has been improved, I believe that it will match all the criteria of excellence of SciPost.

Two minor comments:

* Fig. 1 is of poor quality. Also I recommend to include the figures in the text (they are presently deported after the citation list).

* Appendix A might have been written to avoid a possible negative comparison with BEC physics from an ill-disposed referee. I suggest the authors be less politically correct and clearly stress that (1) in some regimes, one might consider that their optical analogue is a better black-hole analogue than realized in a BEC; and (2) they cannot reach the long wavelength dispersionless thermal limit reached in BECs analogues.

  • validity: high
  • significance: high
  • originality: high
  • clarity: low
  • formatting: below threshold
  • grammar: -

Author:  Maxime Jacquet  on 2020-06-04  [id 847]

(in reply to Report 1 on 2020-02-25)
Category:
answer to question
correction

We thank the referee for their positive comments on the timeliness of this work and the importance of the results relating correlations and entanglement in gravity analogues. We are pleased that they recommend publication of the manuscript in SciPost Physics.

We thank the referee for pointing out the inconsistency in the notation for S. We have now modified this as follows: S(\omega) is introduced at the end of section 2 with the same notation as in Eq.(2). The explicit dependency of all terms on \omega is dropped in Eq.(3) and we comment on this after the equation. With this definition of S in section 2 and its use in section 3, the formulas for the quantum emission are now clear.

We agree with the referee that the discussion of the quantum emission is too short. Thus we have extended section 3 to discuss the statistics of quantum emission: (i) we give the amplitude of significant correlation coefficients; (ii) we discuss the correlation structure when there is no horizon in detail to show that without horizons strong correlations exist between more than two modes; (iii) we comment on the fact that “horizons lead to strong correlations but the converse is not true”. (quoted text can be found verbatim in the manuscript)

Along the lines of the remarks on section 4, we have modified the section to expand the discussion on entanglement.
• in the introductory paragraph, we comment on the mixedness of the state and say that “[...] coupling to modes outside the optical branch is responsible for this”.
• We have moved the content of appendix B to the end of section 4 of the main text and plotted the relation between the correlation strength C and the degree of entanglement J on a Log-Log scale. Interestingly, this reveals two “regimes” of reliability for J given C, which was hardly visible with the linear scale used previously. We comment on this as follows: “States of different degree of mixedness may exhibit the same correlation coefficient. Furthermore, we remark firstly that for a finite correlation coefficient the degree of entanglement is limited below unity. Secondly, correlations close to unity indicate close to maximal entanglement. The degree of entanglement obtained for C<0.5 varies largely, whereas for C>0.5 the degree of entanglement can be reliably inferred from the correlations. Lastly, even small correlation coefficients indicate some entanglement. Further investigations of the relation between J and C beyond this particular analogue are needed.”

The manuscript was submitted as a letter and thus kept short. We have found the referees suggestions very helpful in expanding sections 3 and 4 (see reply to comments above).

On Comment 1: Fig. 1 now has a higher definition. The figures are apended after the citation list because of the submission option of the SciPost class and will appear in the text in the published version.

On Comment 2: We appreciate the suggestion. In App. A, we compare the kinematics of the BEC and optical systems, but do not make a statement about the better analogue. In the main text, we find that the kinematics are responsible for the quantum statistics of the output state. We have rephrased our comment as follows: “If one focuses the analysis on the kinematics of modes u/out and d2/out at low k, the dispersion is approximately linear and the flow velocity is subsonic in the left region and supersonic in the right region. The mode analysis above shows, when input and output modes are taken into account, that two-way motion is possible in both regions of the BEC.” We further stress that point in the conclusion of App. A (by adding “in particular”): “Our analysis of the optical analogue demonstrates that all modes (optical as well as non-optical) significantly modify the quantum state away from the close-to two-mode squeezed vacuum in particular when two-way motion is possible in both regions. Future work should apply this analysis to the BEC analogue also.”

---

## Round 1 · Referee Report · Anonymous (Referee 2) · 2020-2-29

Strengths

Quantum correlations between photons are expected to be a most reliable smoking gun of the quantum nature of the light emitted from the moving refractive index jump, so it is of crucial importance to characterize them in full detail if such systems are to be used as analog models. The authors pinpoint intriguing two- vs. multi-mode features in the correlation pattern at different frequencies.

Weaknesses

The manuscript is very difficult to follow unless one is familiar with the technicalities of the system.
The theoretical description of vacuum emission from moving refractive index jumps is the standard one. The existence of correlations in the quantum emission is known.
The authors do not provide physical insight on the physical mechanisms underlying the observed features, e.g. on the reasons underlying the two vs. multi-mode nature of the correlations.
The title does not really match the content and may be misleading: all the discussion concerns a sudden jump of the refractive index, so there is no real space-time curvature at play, just juxtaposition of two space-time regions that behave differently for waves at different frequencies.

Report

I have two main concerns with this manuscript that keep me from recommending publication.

On one hand, the manuscript does not report any groundbreaking result. Given the structure of the system and the complex wave dynamics, it is natural to expect the presence of complex correlation patterns. Even though this is not necessarily a serious concern against publication, the real issue is that no physical insight is provided on the specific features that are found in the different regimes. In my opinion, this would be the interesting physics that is worth being developed in a research work on this subject and that may justify publication on SciPost.

On the other hand, I find the manuscript very hard to follow unless one is already familiar with the system. This makes the manuscript of limited utility for generic readers. There are a lot of jargon expressions and implicit assumptions, e.g. on the structure and the number of the modes involved in the process, that may mislead the naive reader.

From a general perspective, I personally do not see the reason of publishing the few results on quantum correlations in a separate work rather than merging this manuscript and 1908.02060 into a single and comprehensive long paper (there, the review of the well-known general theory is carried out in a reasonably transparent way). Of course, the situation would dramatically change if the authors were able to expand their discussion of quantum correlations shining important new light on the underlying physics.

Requested changes

See above

  • validity: good
  • significance: ok
  • originality: ok
  • clarity: low
  • formatting: good
  • grammar: good

Author:  Maxime Jacquet  on 2020-06-04  [id 848]

(in reply to Report 2 on 2020-02-29)

We thank the referee for acknowledging the “crucial importance” of photon correlations in optical analogues, as we are investigating these for the first time. We share the view that the two vs. multimode correlations are “intriguing” and not necessarily expected.

We would like to point out that the photon-number correlations are only part of the novelty claim of this submission. We are also investigating the amount of generated entanglement as well as the achieved degree of entanglement. Analysing these three quantities in connection to the single particle spectra we find a connection between the quantum state (beyond photon-number correlations) and the kinematics of modes, or the spacetime curvature, which the analogue is modelling. It is only this connection between our novel observations which makes this manuscript a self-contained work.

We have carefully improved the readability of the manuscript for readers not familiar with the technicalities of the system. Mainly, we have removed the repeated citation of what was reference [1].

It is true that the theoretical description of the index step is taken from literature (there is no “standard” as various authors use different systems, possibly e.g. without other branches in the dispersion relation). Likewise, we are not first to claim correlations in the emission, although the photon number correlations had not been quantified. However, we are not claiming correlations as such as a novelty and are citing previous work in this area.

The referee claims that the authors do not provide “insight on the physical mechanisms”. We would like to reiterate that we find a connection between the quantum state (beyond photon-number correlations) and the kinematics of modes, or the spacetime curvature, which the analogue models. This is indicated in the title and abstract. Clearly also, the theoretical description implies that the continuity of fields across the refractive index front (RIF) induces a classical coupling of fields of opposite norm, leading to a Bogoliubov transformation in the quantum field analysis. Beyond the correlations, the referee does not comment on aspects of the manuscript related to entanglement.

The title of this manuscript therefore is not “misleading”. In a flat spacetime, light propagates on straight lines. The velocity change (group and phase) due to the raised index clearly leads to non-straight propagation of light rays in the geometrical optics approximation. In optical terms, refraction can be continuous, but is still refraction in the discrete case (Snell’s law).

The results of the paper demonstrate how the application of entanglement measures adopted from continuous variable quantum information science to an analogue spacetime curvature leads to dramatically different quantum states, which are complex, but characteristic. This situation arises from quite simple wave kinematics – rather than complex dynamics - , because of the simplicity of a RIF. Physical insight into these quantum states is gained, because they result from the wave kinematics, i.e. the frequency. Within a regime of spacetime curvature, insight is gained by the comparison to the real black hole/ white hole case. A key finding is the dependence of entanglement on the presence of horizons, which gives insight into the entanglement generation. This is not necessarily a “natural expectation”. Also the transition between two- and multimode correlations or the relation between photon number correlations and entanglement could hardly be expected. Questions on physical interpretation of the strength of particular mode couplings are not addressed in the manuscript, although their origin is clearly connected to the matching of fields at the interface in the analytical model. The question of further physical insight is always an interesting one, but certainly not the sole “interesting physics that is worth being developed”.

We do appreciate that the manuscript might be hard to follow. We have implemented a number of clarifications in the text, which raised the accessibility to the required level. In particular, the mode structure is represented by the dispersion relation, which is the generic dispersion relation of optical materials, which we have now explained and referenced. The number of modes is explained now: “The refractive index of most dielectrics is sufficiently well described by 3 resonances.”.

---

## Round 2 · Referee Report · Anonymous (Referee 2) · 2020-7-1

Report

The authors have performed only minor changes to the manuscript in response of my and the other Referee's remarks. Most of my objections still apply to the revised version. From this, I conclude that the authors are not willing to make any serious effort to reinforce the manuscript according to the review reports. This is a bit of a pity as the subject is very interesting and I have the feeling there is a lot of exciting physics remains to be explored at a relatively weak effort. As it is now, the paper reports the finding of an interesting link between different quantities, but does not really illuminate the readers on the underlying mechanisms. Moreover, it is a bit annoying that important details of the section on new results are postponed (see footnote 4) to a forthcoming publication: according to the journal's policy, it should be as self-contained as possible.

Anyway, even though I am not really happy with the manuscript, I have no objections against its publication.

---

## Round 2 · Referee Report · Anonymous (Referee 1) · 2020-7-6

Report

I am surprised by the little care the authors have dedicated to the improvement of their work. Although some of the obvious flaws of the presentation of the first version have been cured, others have been introduced, as the renaming of the axis label of Fig. 6, which makes it more difficult to connect this figure with the quantities defined in the text. Is it ln(I) plotted as a function of ln(C) ?

But my main concern is about secs. 3 and 4. The authors didn't try to improve their very concise presentation. I find the suggestion of the first report of the other referee quite relevant: it would make sense to merge this work with "1908.02060 into a single and comprehensive long paper".

Of course, the author should decide themselves of the publishing policy of their own work. But, as far as the revue is concerned, I do not think that the level of clarity of this manuscript matches the editorial criteria of a journal of the quality of SciPost Physics.

---

## Round 2 · Author Response

We have considered the referee reports and implemented major revisions to our manuscript as described by our replies to the referees.

---

## Round 2 · List of Changes

Abstract: amend final sentence to "The quantum state is a diagnostic for the mode conversion in laboratory tests of
quantum field theory on curved spacetimes."

After Eq.1, insert sentence "The refractive index of most dielectrics is sufficiently well described by 3 resonances."

Final paragraph of section 2, modify $S_\omega$ to $S(\omega)$

After Eq. 3, insert sentence "In (3) the notation is omitting dependencies on $\omega$."

In paragraph discussing Fig. 3 b and d, insert correlation strengths.

New paragraph discussing Fig.3 a, c and e "The correlation structure is very different when there are no horizons. In the low frequency interval (Fig.3 a), significant correlations exist between three mode pairs simultaneously: C(N(nuL),N(uL))=0.56, C(N(noL),N(uoL))=0.97 and C(N(nlL),N(lL))=0.68. In the middle frequency interval (Fig.3 c), significant correlations exist between five modes pairs: C(N(nuL),N(uL))=0.43, C(N(noL),N(uoL))=0.71, C(N(noL),N(moR))=0.67, C(N(nlL),N(loL))=0.47 and C(N(nlL),N(lL))=0.43. In the high frequency interval (Fig.3 e), significant correlations exist between four mode pairs: C(N(nuL),N(uL))=0.83, C(N(noL),N(uL))=0.52, C(N(nlL),N(loL))=0.80, and C(N(nlL),N(lL))=0.50. Without horizons, strong correlations exist albeit with more than two modes."

New penultimate paragraph of Section 3: "In addition to the observations above, we note that strong correlations are possible without horizons (e.g. in Fig.3 a). Therefore, horizons lead to strong correlations but the converse is not true. Clearly, if we considered a two-mode system without horizons, any emission would be in correlated pairs with unit correlation coefficient."

First paragraph of section 4: append ", and coupling to modes outside the optical branch is responsible for this." to final sentence.

Fig 7 is now Fig 6, plot with Log-Log scale.

Discussion of (new) Fig 6 is now at the end of section 4 (content moved from App. B), with the following modification: insert "Furthermore, we remark firstly that for a finite correlation coefficient the degree of entanglement is limited below unity. Secondly, correlations close to unity indicate close to maximal entanglement. The degree of entanglement obtained for C<0.5 varies largely, whereas for C>0.5 the degree of entanglement can be reliably inferred from the correlations. Lastly, even small correlation coefficients indicate some entanglement. Further investigations of the relation between J and C beyond this particular analogue are needed." before final sentence of paragraph.

Appendix A: modify penultimate paragraph to "If one focuses the analysis on the kinematics of modes `u|out' and `d2|out' at low k, the dispersion is approximately linear and the flow velocity is subsonic in the left region and supersonic in the right region. The mode analysis above shows, when input and output modes are taken into account, that two-way motion is possible in both regions of the BEC.".

Appendix A: insert "in particular" in first sentence of final paragraph.

---

## Editorial Decision

published